**Subject Category:**
Biology (whole organism)

health and disease and epidemiology/behaviour

telomere length, telomere attrition, longitudinal, measurement error, regression to the mean, collider bias

**Author for correspondence:**
Melissa Bateson
e-mail: melissa.bateson@ncl.ac.uk

# Controlling for baseline telomere length biases estimates of the rate of telomere attrition

Melissa Bateson[1], Dan T. A. Eisenberg[2]
and Daniel Nettle[1]

[1]Centre for Behaviour and Evolution and Institute of Neuroscience, Newcastle University, Henry Wellcome Building, Framlington Place, Newcastle upon Tyne NE2 4HH, UK
[2]Department of Anthropology, University of Washington, Seattle, WA, USA

MB, 0000-0002-0861-0191; DTAE, 0000-0003-0812-1862; DN, 0000-0001-9089-2599

Longitudinal studies have sought to establish whether environmental exposures such as smoking accelerate the attrition of individuals' telomeres over time. These studies typically control for baseline telomere length (TL) by including it as a covariate in statistical models. However, baseline TL also differs between smokers and non-smokers, and telomere attrition is spuriously linked to baseline TL via measurement error and regression to the mean. Using simulated datasets, we show that controlling for baseline TL overestimates the true effect of smoking on telomere attrition. This bias increases with increasing telomere measurement error and increasing difference in baseline TL between smokers and non-smokers. Using a meta-analysis of longitudinal datasets, we show that as predicted, the estimated difference in telomere attrition between smokers and non-smokers is greater when statistical models control for baseline TL than when they do not, and the size of the discrepancy is positively correlated with measurement error. The bias we describe is not specific to smoking and also applies to other exposures. We conclude that to avoid invalid inference, models of telomere attrition should not control for baseline TL by including it as a covariate. Many claims of accelerated telomere attrition in individuals exposed to adversity need to be re-assessed.

## 1. Introduction

Leucocyte telomere length (LTL)—the mean number of TTAGGG sequence repeats at the end of leucocyte chromosomes—is emerging as a widely studied biomarker of human health. Many cross-sectional studies of LTL demonstrate that mean LTL is

**Box 1.** List of abbreviations.

LTL: leucocyte telomere length

$LTL_b$: True LTL at the baseline time point. Units are bp (base pairs)

$LTL_{fu}$: True LTL at a follow-up time point. Units are bp

$\Delta LTL$: True change in LTL between baseline and follow-up (calculated as $LTL_{fu} - LTL_b$); telomere attrition is thus a negative value of $\Delta LTL$. Units are bp yr$^{-1}$

$mLTL_b$: Measured LTL at the baseline time point. Units are bp

$mLTL_{fu}$: Measured LTL at a follow-up time point. Units are bp

$m\Delta LTL$: Measured change in LTL between baseline and follow-up (calculated as $mLTL_{fu} - mLTL_b$). Units are bp yr$^{-1}$

$error_b$: LTL measurement error at baseline

$error_{fu}$: LTL measurement error at follow-up

CV: Coefficient of variation (standard deviation/mean) of measurement error. Expressed as %

shorter in individuals that have been exposed to diverse forms of adversity [1]. Recent meta-analyses show that LTL tends to be shorter in individuals who are smokers [2,3], are more sedentary [4,5], are obese [6], were subjected to childhood trauma [7] or psycho-social stress [8], suffer from schizophrenia [9,10], post-traumatic stress disorder [11], anxiety or depression [12,13] or have higher perceived stress [14]. These studies have been widely assumed to support the hypothesis that the exposure increases the rate of LTL attrition. However, a cross-sectional association between an exposure and LTL does not necessarily imply a causal link between the exposure and telomere attrition: further evidence for causation is required [15]. A common source of such evidence comes from studies demonstrating that the same exposures associated with shorter LTL cross-sectionally are also associated with faster LTL attrition within individuals over time. To obtain such evidence, telomere attrition is estimated from longitudinal datasets in which LTL is measured at least twice in each individual, first at baseline ($LTL_b$) and again at follow-up ($LTL_{fu}$; see box 1 for abbreviations). The best estimate of the change in telomere length (TL) for a given individual is then simply the difference between the baseline and follow-up measurements ($\Delta LTL$; where negative values indicate telomere attrition). Multiple regression approaches are typically used to estimate the associations between exposure variables and the rate of telomere attrition [16–22]. In the current paper, we address the question of how these statistical models should be constructed in order to obtain unbiased estimates. As we explain below, there are strong theoretical reasons to predict that the current practice of controlling statistically for $LTL_b$ biases estimates of the difference in $\Delta LTL$ between groups of individuals with different exposures and increases the probability of false-positive results. While our discussion is relevant to all of the exposures listed above (and also other factors implicated in accelerated telomere attrition including age [16,18,19,23] and male sex [24]), here, we use the comparison of smokers and non-smokers to illustrate the impact of different analytic strategies.

Researchers often have a strong intuition that it is important to control for baseline variation in the outcome variable of interest in analyses of change. In the current context, this implies including $LTL_b$ as a covariate (i.e. a continuous predictor variable for which a regression coefficient is estimated) in analyses of the association between smoking and $\Delta LTL$ (models 2 and 3 in table 1). We have found 11 studies that report the association between smoking and $\Delta LTL$ and all of these control for $LTL_b$ in their multiple regression models by including it as a covariate [16–21,23,25–28]. What are the arguments in favour of controlling for $LTL_b$ in this way?

In a highly cited paper, Vickers & Altman [29] consider the best analytic approach for controlled trials of an intervention with baseline and follow-up measurement. They show that analysis of covariance (which controls for baseline measurement in an analysis of change) yields the largest estimate (of the models they compared) for the effect of the intervention on the measured outcome variable. They argue that analysis of covariance is generally the most powerful analytic approach, and that the efficiency gains from controlling for baseline will be greatest when the correlation between baseline and follow-up measurements is low. This paper is cited as the justification for controlling for $LTL_b$ in at least one study of the factors associated with $\Delta LTL$ [28]. In studies of telomere dynamics, the correlation between baseline and follow-up telomere measurements is often low (for example, Bendix et al. [16] report a Pearson correlation of only 0.38), apparently providing a strong argument for controlling for $LTL_b$ in analyses of $\Delta LTL$.

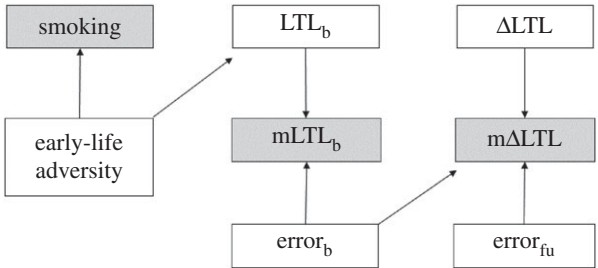

**Figure 1.** Directed acyclic graph summarizing the assumed causal relations between smoking, $mLTL_b$ and $m\Delta LTL$. The graph additionally includes the following unmeasured/latent variables: exposure to early-life adversity, true $LTL_b$, baseline measurement error ($error_b$), true telomere change ($\Delta LTL$) and follow-up measurement error ($error_{fu}$). $Error_b$ and $error_{fu}$ are uncorrelated and independent of LTL and $\Delta LTL$. Causal relationships are indicated by arrows. This diagram is analogous to that presented in Glymour *et al*. ([31]; their fig. 3) and Glymour & Greenland ([30]; their figs 12–14) and can thus be subjected to an identical analysis. See text for further details.

**Table 1.** The four statistical models compared.

| no. | **model** outcome variable | fixed predictor variable(s)[a] | equivalent statistical test |
|---|---|---|---|
| 1 | $m\Delta LTL$ | smoking | two-sample *t*-test or multiple regression[b] |
| 2 | $m\Delta LTL$ | $mLTL_b$ + smoking | analysis of covariance or multiple regression[b] |
| 3 | $mLTL_{fu}$ | $mLTL_b$ + smoking | analysis of covariance or multiple regression[b] |
| 4 | mLTL | time point + smoking + time point × smoking[c] | repeated-measures analysis of variance or mixed-effects model |

[a]Smoking and time point are categorical variables with two levels each (smoker/non-smoker and baseline/follow-up, respectively) and $mLTL_b$ is a continuous variable.
[b]Multiple regression is appropriate if additional control variables are included (e.g. age, sex, race, etc.).
[c]Model 4 additionally contains a random effect (intercept) of participant to account for repeated measures on individuals.

However, although controlling for differences in $LTL_b$ can increase regression coefficients and hence improve statistical power, there is an established epidemiological literature showing that this practice can yield biased estimates and hence spurious false-positive results. One scenario in which bias occurs is when the outcome variable is measured with error [30]. For example, Glymour *et al*. [31] examined the consequences of controlling for baseline cognitive function in asking whether educational attainment affects change in cognitive function in old age. They showed that baseline control induces a spurious statistical association between education and change in cognitive function because of measurement error. More generally, they conclude that when exposures are associated with baseline health status, an estimation bias arises if there is measurement error in health status.

In the case of LTL, two meta-analyses have confirmed that smokers have shorter LTL than non-smokers in cross-sectional datasets [2,3]. Thus, longitudinal datasets are likely to show a baseline association between smoking and LTL. It is also well established that measurement error is a major problem in telomere epidemiology. In large-scale studies, LTL is most commonly measured via a quantitative PCR-based method [32] and less frequently via a more expensive Southern blot-based method [33]. Both methods involve error, and while the magnitude of this error varies between studies, evidence suggests that the Southern blot method is typically more precise, with one comparison estimating the inter-assay coefficient of variation (CV) as 6.45% for qPCR and 1.74% for Southern blot [34]. Much higher reported inter-assay CVs for both methods are not uncommon (e.g. 9.3% for qPCR [21] and 2.8% for Southern blot [35]). Therefore, controlling for $LTL_b$ in analysis of the association between smoking and $\Delta LTL$ appears to meet the criteria for bias identified by Glymour *et al*. [31].

In order to formally establish whether an analysis is likely to be biased, epidemiologists advocate construction of a directed acyclic graph—a diagram representing the causal relationships among a set of variables [30,31,36]. We used this approach to represent one possible hypothesis for the relationships among smoking, $LTL_b$ and $\Delta LTL$. Figure 1 represents the null hypothesis that smoking does not affect

ΔLTL; we assumed, instead, that the association between smoking and $LTL_b$ is brought about by both variables being caused by a third variable. We assumed that this third variable is exposure to early-life adversity [15], but it could equally be a genetic difference. To reflect the presence of error in the measurement of LTL, we distinguish between true and measured values of LTL and ΔLTL; measured values are indicated with a prefix of m. Although we are ultimately interested in true LTL and ΔLTL, these are latent variables that are not directly accessible. Any analysis must therefore use mLTL and mΔLTL. We assume that $mLTL_b$ is positively related to true $LTL_b$ and baseline measurement error ($error_b$), and that mΔLTL is positively related to ΔLTL and follow-up measurement error ($error_{fu}$). However, mΔLTL must also be negatively related to $error_b$ (for a proof, see the electronic supplementary material, equations S1–S4). This is due to regression to the mean: the phenomenon whereby subjects measured with an extreme error, negative or positive, at baseline will on average tend to be measured with a less extreme error at follow-up, generating the negative correlation between $mLTL_b$ and mΔLTL that is commonly observed in longitudinal telomere datasets [37].

In figure 1, a path connects smoking with $mLTL_b$ via early-life adversity and $LTL_b$. Early-life adversity is assumed to cause both smoking and $LTL_b$ (in a directed acyclic graph, a path is a series of lines connecting two variables, regardless of arrow direction). Thus, as long as early-life adversity is not controlled for, a negative association will be present between smoking and $mLTL_b$. A path also connects smoking with mΔLTL via early-life adversity, $LTL_b$, $mLTL_b$ and $error_b$. On this path, $mLTL_b$ is caused by both $LTL_b$ and $error_b$ and is therefore what is termed a 'collider', a common effect of our outcome and predictor variables (mΔLTL and smoking, respectively). In the parlance of directed acyclic graphs, a collider blocks a path, meaning that smoking is independent of mΔLTL under our null hypothesis. However, controlling statistically for $mLTL_b$ unblocks the path between smoking and mΔLTL and hence introduces a spurious association between smoking and mΔLTL. This latter phenomenon is known as 'collider bias' [38,39]. In summary, it follows from the assumptions embodied in figure 1 that controlling for $mLTL_b$ should inflate estimates of the association between smoking and mΔLTL via collider bias. The size of this bias should depend on both the presence of an association between smoking and $LTL_b$ and the size of the LTL measurement error.

In the remainder of this paper, we test the above predictions with two complementary approaches. First, we use a simulation model to show numerically that controlling for $mLTL_b$ biases estimates of the association between smoking and mΔLTL and that the size of the bias depends on the size of the LTL measurement error. By using realistic values in our simulation, we determine the likely importance of any bias. Second, we use meta-analysis of seven previously published empirical datasets to test the major assumptions and predictions of our simulation model in real LTL data.

# 2. Simulation model

The advantage of a simulation approach is that it is possible to generate datasets for which the true values of latent variables (in this case, $LTL_b$ and ΔLTL) are known. We can then verify how adding different magnitudes of measurement error and using different statistical models affect estimates of the difference in ΔLTL between smokers and non-smokers. We simulated longitudinal LTL datasets in which we set the true differences between smokers and non-smokers in $LTL_b$, ΔLTL and the LTL measurement error ($error_b$ and $error_{fu}$) based on realistic values obtained from the literature. We then used these simulated datasets to calculate the size of biases in the estimates for the difference in ΔLTL between smokers and non-smokers obtained from different statistical models in which we varied whether we controlled for $LTL_b$.

We compared the four statistical models given in table 1. Model 1 is the basic model in which mΔLTL is predicted by smoking status with no statistical control for $mLTL_b$. Model 1 is rarely found in the telomere epidemiology literature, but is sometimes seen in the analysis of randomized controlled trials of interventions such as physical exercise (e.g. [40]). Model 2 includes control for $mLTL_b$ by adding $mLTL_b$ as a covariate. Model 2 represents the approach recommended by Vickers & Altman [29] and most commonly adopted in the current telomere epidemiology literature (e.g. [17–23,25–27]). Model 3 is a less common variant of model 2 in which the outcome variable is $mLTL_{fu}$ as opposed to mΔLTL (e.g. [16,28,41,42]). Model 4 is a repeated-measures equivalent of model 1 in which the outcome variable is mLTL and time point (baseline versus follow-up) is entered as a categorical predictor (e.g. [43]); in this model, inclusion of the interaction between time point and smoking is necessary to test the hypothesis that mΔLTL differs between smokers and non-smokers. Note that models 1 and 4 contain no control for $mLTL_b$, in that $mLTL_b$ is not included on the right-hand side of the model

**Table 2.** Parameter values used in the simulations.

| | | scenario | | | |
| | | A | B | C | D |
| | | no diff. in $LTL_b$ | | true diff. in $LTL_b$ | |
| | parameter | no diff. in $\Delta LTL$ | true diff. in $\Delta LTL$ | no diff. in $\Delta LTL$ | true diff. in $\Delta LTL$ |
| non-smokers | $LTL_b$ (bp; mean ± s.d.[a]) | 7430 ± 777 | 7430 ± 777 | 7500 ± 777 | 7500 ± 777 |
| | $\Delta LTL$ (bp yr$^{-1}$; mean ± s.d.[a]) | −40.7 ± 46 | −40 ± 46 | −40.7 ± 46 | −40 ± 46 |
| smokers | $LTL_b$ (bp; mean ± s.d.[a]) | 7430 ± 777 | 7430 ± 777 | 7359 ± 777 | 7359 ± 777 |
| | $\Delta LTL$ (bp yr$^{-1}$; mean ± s.d.[a]) | −40.7 ± 46 | −42 ± 46 | −40.7 ± 46 | −42 ± 46 |

[a]Note that these standard deviations of $LTL_b$ and annual attrition are likely to be overestimates of the true values, since both true variation and measurement error contribute to the measured values. However, in the absence of error-free measurements, we used these published standard deviations as the best estimates available.

equation, whereas models 2 and 3 control for $mLTL_b$ by including it as a covariate and estimating its regression coefficient.

## 2.1. Methods

We simulated LTL datasets under four different scenarios for the true differences in $LTL_b$ and $\Delta LTL$ between smokers and non-smokers: (A) no difference in $LTL_b$ and no difference in $\Delta LTL$; (B) no difference in $LTL_b$, but a true difference in $\Delta LTL$; (C) a true difference in $LTL_b$, but no difference in $\Delta LTL$; and (D) a true difference in $LTL_b$ and a true difference in $\Delta LTL$ (table 2). The parameter values used in each scenario were taken from Aviv *et al.* [26], who report a small, but significant, difference in $mLTL_b$ between smokers and non-smokers of 141 bp and a non-significant $m\Delta LTL$ between smokers and non-smokers of −2 bp yr$^{-1}$. We chose this study because LTL was measured using Southern blot and the reported inter-assay CV is only 1.4%. Thus, the LTL measurements are likely to be reasonable estimates of the true values.

The simulation of LTL values was implemented in the statistical computing language R. In each replicate simulation, values of $LTL_b$ were generated for 2000 participants (1000 non-smokers and 1000 smokers) by drawing independent random samples from normal distributions with means and standard deviations given in table 2. Each participant was then assigned a value of $\Delta LTL$ year$^{-1}$ by again drawing an independent random sample from normal distributions for $\Delta LTL$ with means and standard deviations given in table 2. This rate of change was applied for 10 years starting with the true $LTL_b$ to yield a true $LTL_{fu}$ for each participant. We assumed that each participant experienced a constant value of $\Delta LTL$ over the follow-up interval. Measurement error was introduced into both $LTL_b$ and $LTL_{fu}$ by assuming that mLTL was an independent random sample from a normal distribution with the mean equal to the true LTL and the standard deviation equal to the true $LTL \times CV/100$, where CV is the coefficient of variation of the measurement error expressed as a percentage. Measured $\Delta LTL$ for each participant was calculated as the difference between $mLTL_b$ and $mLTL_{fu}$. We assumed values of CV of 0, 1, 2, 4, 8 and 16%, and generated 1000 replicate datasets for each value of CV in each of the four scenarios (A, B, C and D). Note that while these CV values describe various levels of measurement error within our simulations, these specific CV values cannot be straightforwardly compared to the CVs from laboratory measures reported in empirical papers due to varying zero-points (see [44] for discussion of the comparability of CVs).

We modelled the dataset from each replicate with the four different models summarized in table 1. Models 1–3 are variants of the general linear model and were fitted using the 'lm' function in the R base package, whereas model 4 is a general linear mixed-effects model and was fitted using the 'lmer' function in the R package 'lme4' [45].

To compare the estimates of the difference in $m\Delta LTL$ between smokers and non-smokers produced by the different models, we extracted the $\beta$ coefficients for the 'Smoking' variable produced by models 1–3 and the 'time point × smoking' variable for model 4. To analyse type 1 errors (the probability of

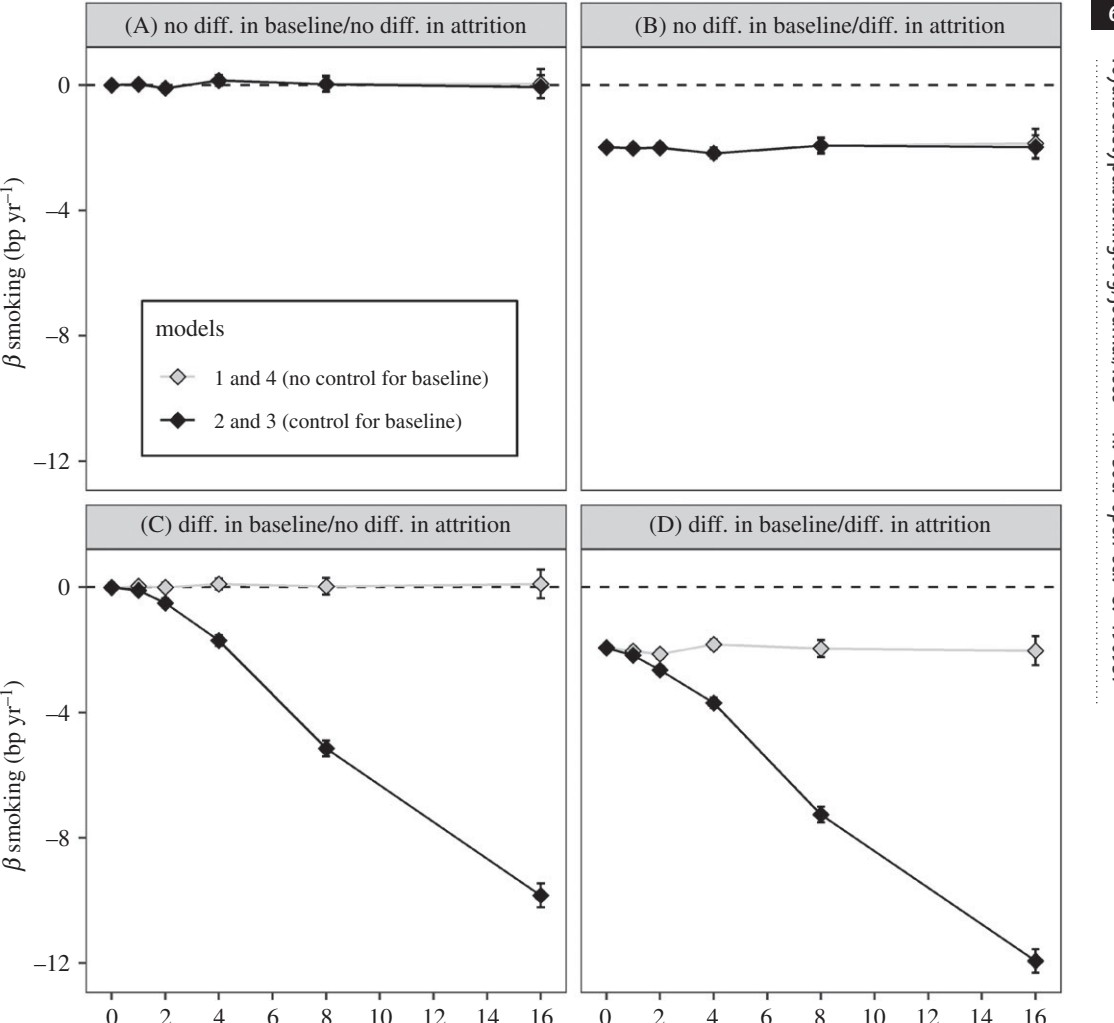

**Figure 2.** Controlling for LTL$_b$ exaggerates estimates of the difference in ΔLTL between smokers and non-smokers when there is a difference in LTL$_b$. The estimated difference in mΔLTL between smokers and non-smokers as a function of measurement error (CV). The $\beta$ estimates were obtained by fitting four alternative models to data simulated, given four sets of assumptions regarding the true differences between smokers and non-smokers (scenarios A–D in table 2). The dashed lines indicate no difference in mΔLTL between smokers and non-smokers. Data points are the mean ± 95% confidence intervals obtained from modelling the data from 1000 replicate simulations. The four scenarios were as follows: (A) no difference in LTL$_b$ and no difference in ΔLTL; (B) no difference in LTL$_b$ but a true difference in ΔLTL; (C) a true difference in LTL$_b$ but no difference in ΔLTL; and (D) a true difference in LTL$_b$ and a true difference in ΔLTL. The true difference in LTL$_b$ between smokers and non-smokers in scenarios C and D was LTL$_b$ 141 bp shorter in smokers. The true difference in ΔLTL between smokers and non-smokers in scenarios B and D was ΔLTL −2 bp yr$^{-1}$ faster in smokers.

incorrectly rejecting the null hypothesis of no difference in ΔLTL between smokers and non-smokers in scenarios where there was no true difference) and statistical power (the probability of correctly rejecting the null hypothesis of no difference in ΔLTL in scenarios where there was a true difference), we additionally recorded whether the $\beta$ coefficient was significantly different from zero (at $p < 0.05$ as widely employed) in each analysis. Summarized output from one run of the simulation is available at the following doi:10.5281/zenodo.1009086. These data were used to create figures 2 and 3 and electronic supplementary material, figure S1.

To test the sensitivity of our results to various assumptions, we conducted the following additional simulations. First, to examine sensitivity to the size of the difference in LTL$_b$ between smokers and non-smokers in scenarios C and D, we re-ran the simulation with differences of: 0, 100, 200, 400, 800 and 1600 bp. (Our rationale for including differences up to 1600 bp was that assuming age-related attrition of 40 bp yr$^{-1}$, a 1600 bp difference would be expected between 20 and 60 year olds, meaning that for

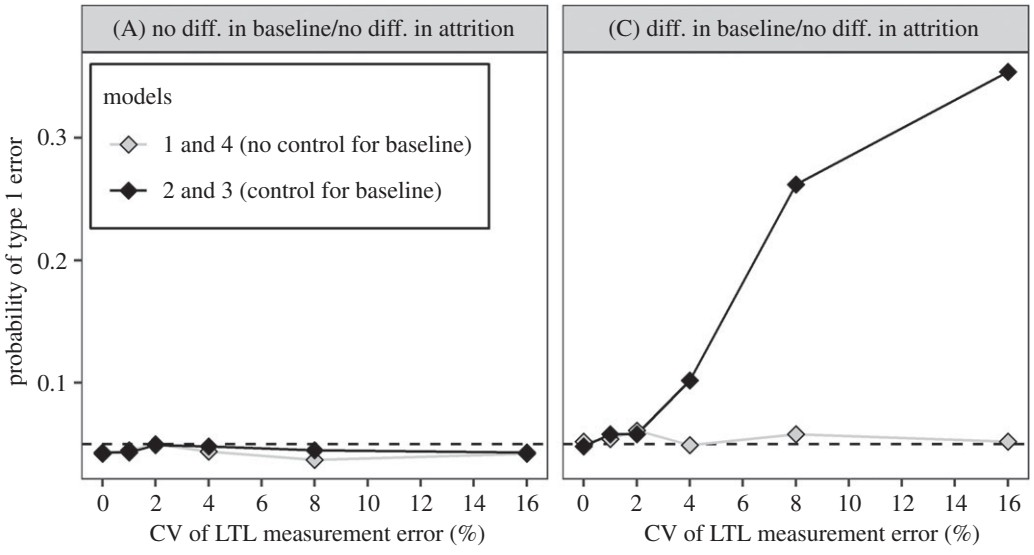

**Figure 3.** Controlling for $LTL_b$ increases the probability of false-positive errors when there is a difference in $LTL_b$. Probability of a type 1 error as a function of measurement error (CV) for the four models under consideration. Data points represent the proportion of simulations yielding a *p*-value below 0.05 in 1000 replicate simulations. The left and right panels show the probability of type 1 errors in scenarios A and C, respectively. The difference in $LTL_b$ between smokers and non-smokers in scenario C was $LTL_b$ 141 bp shorter in smokers.

analyses of the effect of age on $\Delta LTL$ year$^{-1}$, this value would be realistic.) Second, to examine sensitivity to the size of the study, we re-ran the simulation with the following numbers of participants (half smokers and half non-smokers): 200, 400, 800, 1600, 3200 and 6400. Third, to examine sensitivity to the true difference in $\Delta LTL$ between smokers and non-smokers in scenarios B and D, we re-ran the simulation with a true difference of $-20$ bp yr$^{-1}$ ($\Delta LTL$ of $-50$ bp yr$^{-1}$ in smokers and $-30$ bp yr$^{-1}$ in non-smokers). Fourth, to examine sensitivity to the assumption that LTL measurement error is proportional to LTL, we re-ran the simulation with non-proportional measurement error. We used the following standard deviation values to calculate the measurement error: 0, 70, 140, 280, 560 and 1120 bp.

Finally, we explored the impact of correcting $m\Delta LTL$ for regression to the mean caused by measurement error prior to fitting statistical models. We used the equation suggested by Verhulst *et al.* [37] to compute $D$ (see electronic supplementary material, equation S5) and re-ran the statistical models with this new outcome variable in place of $m\Delta LTL$.

## 2.2. Results

### 2.2.1. Accuracy of parameter estimates

In scenario A, in which there is no difference in either $LTL_b$ or $\Delta LTL$ between smokers and non-smokers, all models correctly estimate the true difference in $\Delta LTL$ as zero (figure 2 scenario A). However, in scenario C, in which there is a difference in $LTL_b$, but no difference in $m\Delta LTL$, while models 1 and 4 correctly estimate the difference in $\Delta LTL$ as zero, models 2 and 3 overestimate it at non-zero values of measurement error, and this overestimation increases as LTL measurement error increases (figure 2 scenario C). In scenario B, in which there is no difference in $LTL_b$, but a true difference in $\Delta LTL$, all models correctly estimate the difference in $\Delta LTL$ at around $-2$ bp yr$^{-1}$ (figure 2 scenario B). However, in scenario D, in which there is a difference in $LTL_b$ and a true difference in $\Delta LTL$ of $-2$ bp yr$^{-1}$, while models 1 and 4 correctly estimate the difference in $\Delta LTL$, models 2 and 3 overestimate it at non-zero values of measurement error, and this overestimation increases as measurement error increases (figure 2 scenario D). The magnitude of the bias produced by models 2 and 3 in scenarios C and D is the same, and is hence independent of the presence of a true difference in $\Delta LTL$.

### 2.2.2. Type 1 error rate and power

In scenario A, the probability of type 1 errors based on a sample size of 2000 is around 0.05 for all models (figure 3 scenario A). However, in scenario C, the type 1 error rates for models 2 and 3 are

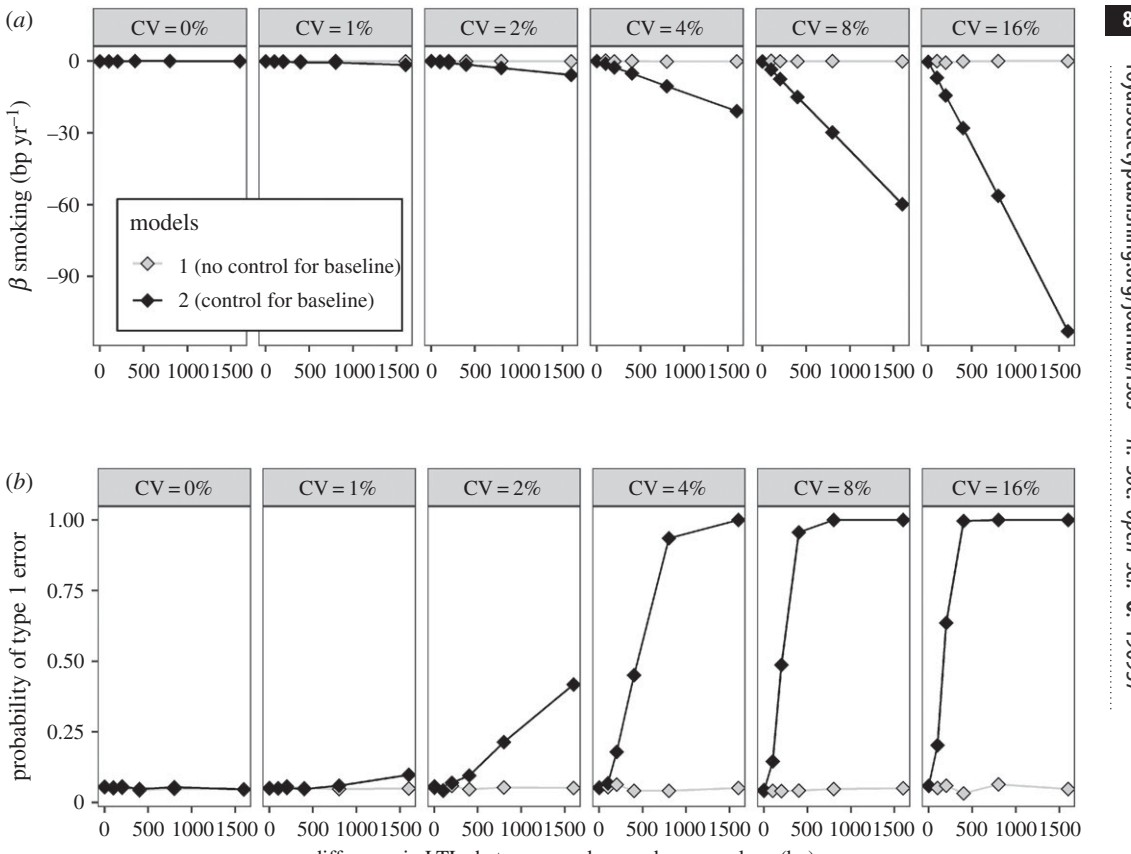

**Figure 4.** The bias caused by controlling for LTL$_b$ is a synergistic interaction between difference in LTL$_b$ and measurement error. The data in this figure come from a simulation of scenario C only (a true difference in LTL$_b$, but no difference in ΔLTL). (a) The estimated difference in mΔLTL between smokers and non-smokers as a function of the difference in LTL$_b$ and CV for models 1 and 2. Data points are the mean ± 95% confidence intervals obtained from modelling the data from 1000 replicate simulations. (b) The probability of a type 1 error as a function of the difference in LTL$_b$ and CV for models 1 and 2. Data points represent the proportion of simulations yielding a $p$-value below 0.05 in 1000 replicate simulations.

greater than 0.05 and rise as CV increases (figure 3 scenario C), reflecting the exaggerated estimates of difference in ΔLTL seen in figure 2 scenario C.

In scenario B, the power to correctly reject the null hypothesis of no difference in ΔLTL based on a sample size of 2000 is approximately the same for all models and decreases with increasing CV (electronic supplementary material, figure S1B). The low power reflects the small true effect size of only −2 bp yr$^{-1}$. In scenario D, the power of models 1 and 4 decreases with increasing CV, but the power of models 2 and 3 increases with increasing CV, reflecting the exaggerated estimates of difference in ΔLTL seen in figure 2 scenario D (electronic supplementary material, figure S1D).

The results obtained from models 1 and 4 were identical to each other and different from models 2 and 3 which were identical to each other. Thus, the models fell into two groups determined by whether or not they control for mLTL$_b$. Since models 3 and 4 are redundant, henceforth, we only describe results for models 1 (no control for LTL$_b$) and 2 (control for LTL$_b$).

### 2.2.3. Sensitivity analyses

Varying the difference in LTL$_b$ between smokers and non-smokers in scenarios C and D confirmed that there is a synergistic interaction between difference in LTL$_b$ and CV on the size of the bias arising from model 2 (figure 4a). At high, but realistic, values of the difference in LTL$_b$ and CV, the bias led to near-certain type 1 errors in scenario C (figure 4b).

Varying the numbers of participants in the simulation had no impact on the accuracy of the parameter estimates: biases in scenarios C and D were identical to those seen in figure 2 at all sample sizes (electronic supplementary material, figure S2). Increasing sample size had no impact on the

probability of type 1 errors in scenario A, but increased the probability of type 1 errors with model 2 in scenario C (electronic supplementary material, figure S3) due to the impact of sample size on the $p$-value. For the same reason, increasing sample size increased the power to reject the null hypothesis in scenarios B and D. This increase in power was greater with model 2 in scenario D due to the exaggerated parameter estimates (electronic supplementary material, figure S4).

Increasing the true difference in ΔLTL from −2 to −20 bp yr$^{-1}$ in scenarios B and D had no impact on the size of the biases observed: the difference between the parameter estimates for models 1 and 2 was the same as that seen in figure 2 (electronic supplementary material, figure S5). Concomitantly, there was no impact on the probability of type 1 errors (electronic supplementary material, figure S6). Model 1 correctly estimates the difference in ΔLTL at around −20 bp yr$^{-1}$ in scenarios B and D (electronic supplementary material, figure S5). The larger true effect size results in a huge increase in power in scenarios B and D compared to that seen in electronic supplementary material, figure S1 (electronic supplementary material, figure S7).

Changing the way in which we implemented measurement error from error that was proportional to LTL to non-proportional error had no impact on the size of the biases observed in scenarios C and D (electronic supplementary material, figure S8), the probability of type 1 errors in scenarios A and C (electronic supplementary material, figure S9) or power in scenarios B and D (electronic supplementary material, figure S10).

### 2.2.4. Effect of correcting for regression to the mean

Using $D$ in place of mΔLTL as the outcome variable had no impact on the parameter estimates derived from model 1 (all were accurate), but the parameter estimates derived from model 2 in scenarios C and D were still biased, albeit with a different pattern. Overall, the bias with model 2 was of a smaller magnitude and in the opposite direction compared to when we modelled uncorrected mΔLTL. At low values of measurement error, smokers were incorrectly estimated to have slower telomere attrition than non-smokers in scenarios C and D and this bias declined towards zero as measurement error increased (electronic supplementary material, figure S11C and D).

## 2.3. Discussion

As long as there was no true difference in baseline LTL$_b$ between smokers and non-smokers, then all of the statistical models that we applied accurately estimated the difference in ΔLTL between smokers and non-smokers. However, if there was even a small difference in LTL$_b$ between smokers and non-smokers *and* LTL measurement error was non-zero, then controlling for LTL$_b$ biased estimates of the difference in ΔLTL between smokers and non-smokers. Specifically, the difference in ΔLTL was overestimated and the size of this overestimation increased synergistically with increases in the difference in LTL$_b$ and in LTL measurement error. This bias translated into a type 1 (i.e. false-positive) error rate of above the usually accepted 5% level when there was no true difference in ΔLTL. This rise in the false-positive error rate was exacerbated in studies with larger numbers of participants due to the positive impact of sample size on power. The apparent improvement in power provided by models 2 and 3 in scenario D, seen in electronic supplementary material, figures S1, S4, S7 and S10, and noted by Vickers & Altman [29], is an artefact of biased parameter estimates. Correcting mΔLTL for regression to the mean caused by measurement error using the equation suggested by Verhulst *et al.* [37] does not solve the problem of biased parameter estimates when LTL$_b$ differs between smokers and non-smokers.

It is worth pointing out that scenario B is unlikely to be very common, unless LTL$_b$ is measured early in life, before the participants have started smoking. Likewise, scenario A is not typical, given the abundant cross-sectional evidence that smokers have shorter telomeres than non-smokers [2,3]. Thus, the scenarios likely to be empirically widespread are exactly those (C and D) where bias will occur if LTL$_b$ is controlled for.

We parametrized our simulation for a comparison of smokers and non-smokers. However, for variables where the difference in LTL$_b$ is larger than 141 bp, as could be the case for a comparison of different ages or races, our simulations suggest that false-positive error rates for associations with ΔLTL could approach 100% if LTL$_b$ is controlled for (figure 4).

In conclusion, given that LTL measurement error is never zero, our simulations suggest that models of types 2 and 3, which control for LTL$_b$ by including it as a covariate, should be avoided in the analysis of factors associated with ΔLTL. By contrast, models 1 and 4 yield accurate parameter estimates. Models 1

and 4 yield equivalent results with two telomere measurements, but model 4 is more flexible if, for example, more than two telomere measurements are available.

# 3. Meta-analysis of empirical datasets

On the basis of our simulations, we predict that in real longitudinal datasets, estimates of the difference in ΔLTL between smokers and non-smokers will depend on both the size of the measurement error and the modelling strategy adopted. Specifically, we predict that estimates of the difference in ΔLTL between smokers and non-smokers will be larger when they are derived from models controlling for $mLTL_b$, and that the size of this effect of modelling strategy will increase as measurement error increases.

Here, we test these predictions using real data from seven published longitudinal cohorts. Our specific aims were as follows. First, we set out to confirm that there is substantial variation in LTL measurement error among the seven cohorts. Second, we tested whether the estimated association between smoking and mΔLTL is greater when the association is derived from a model controlling for $LTL_b$ (model 2; table 1) compared with a model without control for $LTL_b$ (model 1), and whether any discrepancy is explained by differences in LTL measurement error among cohorts.

## 3.1. Methods

We used data from participants in seven longitudinal cohorts whose LTL had been measured at least twice and for which data on smoking status were also available (table 3). We restricted our analyses to those participants who were either current or never smokers at the time of the baseline LTL measurement (designated 'smokers' and 'non-smokers', respectively); those who had quit smoking at some point prior to the baseline measurement were excluded.

The first telomere measurement for each participant was designated as $mLTL_b$ and the second, or last where more than two were available (both the Lothian cohorts), as $mLTL_{fu}$. For each participant ΔLTL year$^{-1}$ was calculated as $(mLTL_{fu} - mLTL_b)/(age_{fu} - age_b)$, so that negative values indicate telomere attrition.

To characterize the LTL measurement error present in each cohort, we did not use the CVs reported for the cohorts, because CV values are often incomparable across studies [44]. Instead, we used signatures of measurement error that can be directly calculated from the telomere measurements themselves, namely the correlation between $mLTL_b$ and $mLTL_{fu}$ and the correlation between $mLTL_b$ and mΔLTL [49]. All else being equal, the correlation between $mLTL_b$ and $mLTL_{fu}$ will be weaker the higher the measurement error, and the correlation between $mLTL_b$ and mΔLTL will be more negative the higher the measurement error [37,49].

For each cohort, we modelled the difference in mΔLTL year$^{-1}$ between smokers and non-smokers using models 1 and 2 (table 1). These models yielded estimates of the standardized $\beta$ coefficient for the association between smoking and mΔLTL year$^{-1}$. To compare the difference in the estimates of this parameter between models 1 and 2, we calculated the difference in association ($\Delta\beta = \beta_{model\ 2} - \beta_{model\ 1}$). A more negative association between smoking and mΔLTL year$^{-1}$ in model 2 compared to model 1 will therefore be indicated by a more negative value of $\Delta\beta$. To compare the results obtained across the seven cohorts, we used meta-regression, fitting linear regression models to the values obtained for each cohort weighting data points by the number of participants in each cohort.

## 3.2. Results

### 3.2.1. Descriptive statistics

The combined dataset included data from 1768 adults, comprising 550 current smokers and 1218 never smokers at the baseline measurement. The mean age at baseline of the cohorts was $65.9 \pm 8.5$ years (mean ± s.d.; range: 53.4–80.2) and the mean follow-up interval was $8.5 \pm 1.2$ years (mean ± s.d.; range: 6.0–9.5).

Five cohorts measured LTL using the qPCR method and two used the Southern blot method. For all cohorts, the slope of the regression of $mLTL_{fu}$ on $mLTL_b$ is less than 1 (figure 5a). However, the strength of the relationship differs markedly between cohorts, with the Pearson correlation coefficients ranging from −0.01 to 0.97 (table 3). For all cohorts, the slope of the regression of mΔLTL year$^{-1}$ on $mLTL_b$ is negative (figure 5b). There is a positive association between the correlation coefficient arising from the association

**Table 3.** Summary of the datasets analysed.

| cohort (acronym) | country | mean age at baseline (years) | mean follow-up interval (years) | LTL measurement method | number of participants by baseline smoking status[a] | | diff. in $LTL_b$ between smokers and never smokers (Cohen's $d$)[b] | signatures of LTL measurement error (data from smokers and never smokers pooled) | | diff. in $\Delta LTL$ year$^{-1}$ between smokers and never smokers (standardized $\beta$ (s.e.))[d] | | reference for cohort |
|---|---|---|---|---|---|---|---|---|---|---|---|---|
| | | | | | current smokers | never smokers | | correlation between $LTL_b$ and $LTL_{fu}$ ($r$) | correlation between $LTL_b$ and $\Delta LTL$ ($r$)[c] | model 1[e] | model 2 | |
| ADELAHYDE (ADE) | France | 68.1 | 8.3 | Southern blot | 5 | 42 | −0.99 | 0.93 | −0.09 | 0.49 [0.47] | 0.49 [0.50] | [46] |
| Caerphilly Cohort Study (CCS) | Wales, UK | 64.2 | 8.0 | qPCR | 207 | 169 | −0.12 | 0.03 | −0.81 | 0.22 [0.10] | 0.12 [0.06] | [47] |
| Evolution de la Rigidité Artérielle (ERA) | France | 58.6 | 9.5 | Southern blot | 27 | 86 | 0.19 | 0.96 | −0.32 | −0.30 [0.22] | −0.24 [0.21] | [27] |
| Hertfordshire Ageing Study (HAS) | England, UK | 67.0 | 9.2 | qPCR | 29 | 93 | −0.19 | −0.10 | −0.75 | −0.12 [0.21] | −0.27 [0.14] | [47] |
| Lothian Birth Cohort 1921 (LBC1921) | Scotland, UK | 80.2 | 9.2 | qPCR | 3 | 78 | −0.40 | 0.35 | −0.23 | 0.10 [0.59] | 0.06 [0.59] | [47] |
| Lothian Birth Cohort 1936 (LBC1936) | Scotland, UK | 69.6 | 6.0 | qPCR | 75 | 415 | −0.16 | 0.54 | −0.31 | −0.10 [0.13] | −0.15 [0.12] | [48] |
| MRC National Survey of Health and Development (NSHD) | England, UK | 53.4 | 9.3 | qPCR | 204 | 335 | −0.06 | 0.08 | −0.80 | 0.03 [0.09] | −0.02 [0.05] | [47] |

[a]These numbers are smaller than the numbers given in the original reference for the cohort because we only included participants for whom there was TL and age at both baseline and follow-up and smoking status at baseline; furthermore, participants who had quit smoking prior to baseline were excluded.

[b]Negative numbers indicate that $LTL_b$ is shorter in smokers.

[c]Negative numbers indicate that longer $LTL_b$ is associated with greater telomere loss.

[d]Negative numbers indicate greater telomere loss in smokers.

[e]Models 1 and 2 correspond to models 1 and 2 in table 1.

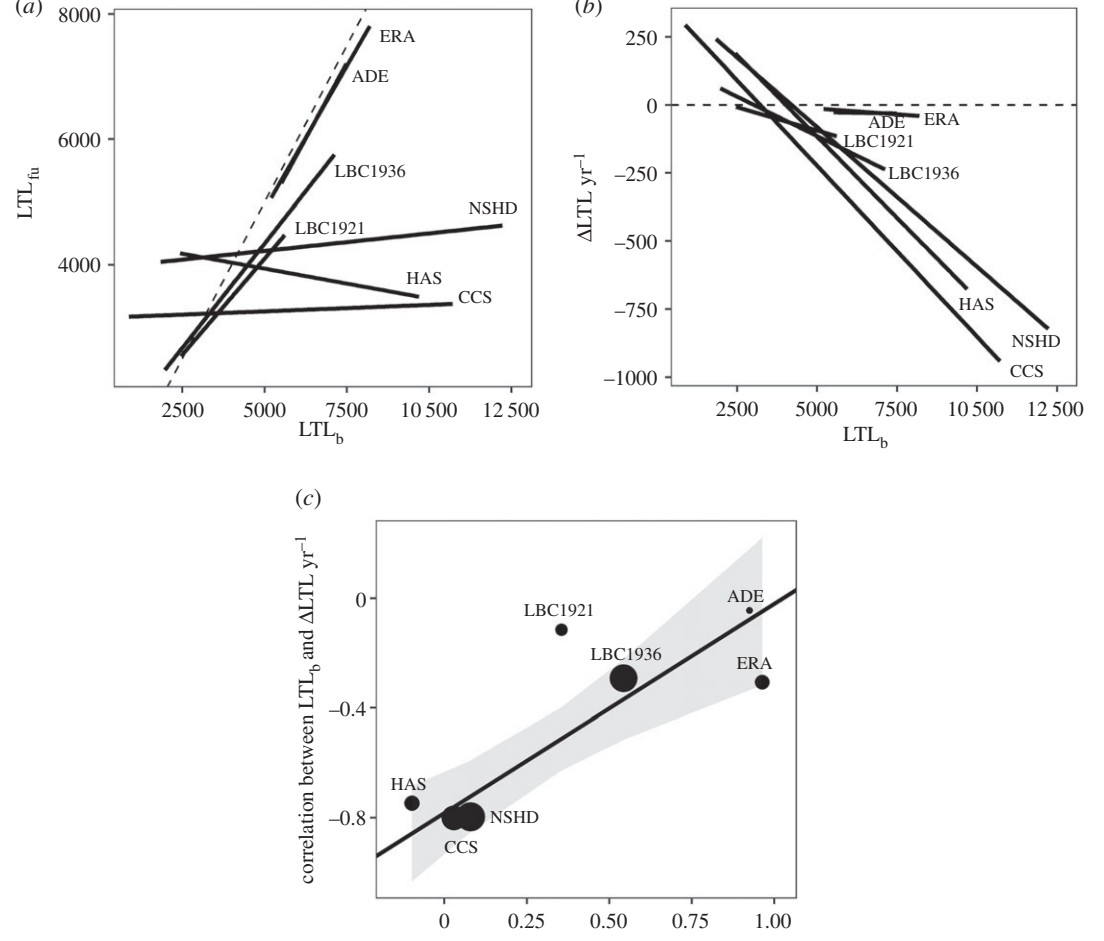

**Figure 5.** Signatures of measurement error differ between cohorts. (*a*) The relationship between mLTL$_b$ and mLTL$_{fu}$ for each of the seven cohorts. The lines were obtained from simple linear regression. The dashed line shows the expectation if there is no change in mLTL between baseline and follow-up. Most of the data fall below the dashed line, indicating that in most participants, mLTL shortened between baseline and follow-up. Slopes closer to one indicate lower measurement error. (*b*) The relationship between mLTL$_b$ and mΔLTL year$^{-1}$ for each of the seven cohorts. The lines were obtained from simple linear regression. The dashed line shows the expectation if there is no measurement error. Flatter slopes indicate lower measurement error. (*c*) Meta-regression between the correlation coefficients derived from the associations shown in (*a,b*). The size of the point representing each cohort is proportional to the number of participants. The solid black line was derived from a linear regression in which the points were weighted by the number of participants in each cohort and the grey ribbon shows the 95% confidence interval for this line. More positive values on both axes correspond to lower measurement error.

between mLTL$_b$ and mLTL$_{fu}$ and the correlation coefficient arising from the association between mLTL$_b$ and mΔLTL year$^{-1}$ (weighted linear regression: $\beta \pm$ s.e. $= 0.76 \pm 0.18$, $t = 4.17$, $p = 0.0088$; figure 5*c*).

### 3.2.2. Effects of modelling strategy

We compared estimates (standardized $\beta$ coefficients) of the difference in mΔLTL year$^{-1}$ between smokers and non-smokers derived from models 1 and 2 (table 3). Coefficients from models 1 and 2 are strongly positively correlated, but not identical (figure 6*a*; weighted linear regression: $\beta \pm$ s.e. $= 0.89 \pm 0.11$, $t = 8.15$, $p = 0.0005$). There is a tendency for the coefficients from model 2 to be more negative, indicating a bigger estimated difference in mΔLTL year$^{-1}$ compared to model 1 (model 1: mean $= 0.046$, s.d. $= 0.258$; model 2: mean $= -0.001$, s.d. $= 0.262$; paired $t$-test: $t_6 = 1.87$, $p = 0.1106$). This difference is greater if the comparison is restricted to the five cohorts measured with qPCR (model 1: mean $= 0.026$, s.d. $= 0.142$; model 2: mean $= -0.052$, s.d. $= 0.158$; paired $t$-test: $t_4 = 3.87$, $p = 0.0180$). There is a positive relationship between the correlation coefficient arising from the association between mLTL$_b$ and mLTL$_{fu}$ (a proxy for measurement error in the cohort) and Δ$\beta$ (a measure of likely bias; weighted linear regression $\beta \pm$ s.e. $= 0.11 \pm 0.04$, $t = 2.91$, $p = 0.0336$; figure 6*b*).

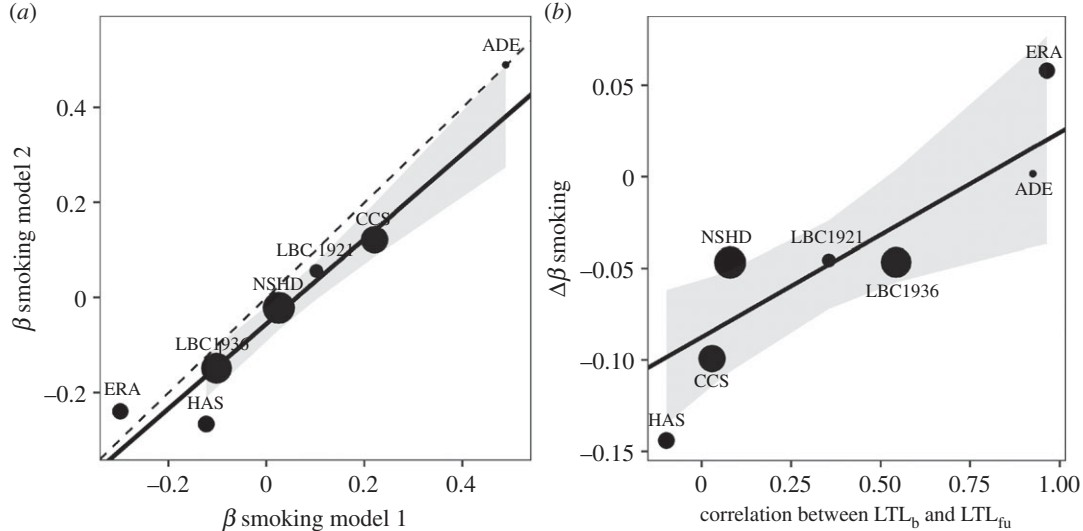

**Figure 6.** The biasing effect of controlling for LTL$_b$. (*a*) The relationship between the $\beta$ coefficients for smoking derived from models 1 (no control for LTL$_b$) and 2 (control for LTL$_b$). The dotted line shows the expectation if the coefficients were identical. (*b*) The correlation between a signature of LTL measurement error (the correlation between LTL$_b$ and LTL$_{fu}$; larger values indicate lower measurement error) and the difference between the $\beta$ coefficients derived from models 1 and 2. In both panels, the solid black line was derived from a linear regression in which the points were weighted by the number of participants in each cohort and the grey ribbon shows the 95% confidence interval for this line.

## 3.3. Discussion

Two proxies for LTL measurement error [49] varied among the seven cohorts: there was variation in both the correlation between mLTL$_b$ and mLTL$_{fu}$ and the correlation between mLTL$_b$ and m$\Delta$LTL. Furthermore, these two proxies were correlated with each other as would be expected if they both reflect measurement error. While we appreciate that there is some evidence that long telomeres may really shorten faster [50], there is no reason to expect that any such biological effect will differ between human cohorts. By contrast, there is good reason to expect that there will be differences in telomere measurement error between human cohort studies. Therefore, it is a reasonable assumption that variation between cohorts in the above correlations reflects variation in measurement error.

When we estimated the difference in $\Delta$LTL between smokers and non-smokers using two modelling strategies, model 1 (no baseline control) and model 2 (baseline control) produced different results: estimates derived from model 2 showed a more negative effect of smoking than those derived from model 1. Since there can only be one true difference in $\Delta$LTL, the estimates derived from either model 1 or model 2 (or both) must be incorrect. The fact that controlling for LTL$_b$ increases estimates of the effect of smoking rather than decreasing them suggests that LTL$_b$ is not a proxy for positive confounders of the difference in $\Delta$LTL between smokers and non-smokers, but instead introduces a bias. Indeed, the directed acyclic graph and simulation analyses both argue that controlling for LTL$_b$ (model 2) yields biased estimates. Thus, it seems likely that model 2 is biased. This conclusion is strengthened by our finding that the size of the discrepancy between the estimates derived from models 1 and 2 is predicted by a proxy for the magnitude of the LTL measurement error present in the cohort.

We do not report the statistical significance of the associations in table 3. Our rationale was that the cohorts are small (47–539 participants) and the majority of the differences were therefore not significant. However, for the cohorts with indications of high measurement error, the likely bias arising from model 2 is sufficient to cause concerns over inference, especially if the studies were larger. For example, in the Hertfordshire Ageing Study, which has a baseline difference of −0.19 s.d. and massive measurement error, the $\beta$ coefficient for the difference in attrition from model 2 (likely biased) are more than double what it is for model 1 (unbiased).

In electronic supplementary material, table S1 and figure S13, we show, using the same datasets, that the above results for smoking generalize to two other variables, sex and body mass index, that are also associated with LTL in cross-sectional studies and have been suggested to cause differences in LTL attrition [6,24]. Thus, controlling for baseline TL in estimating the effect of BMI or sex on telomere attrition leads to a larger estimated effect compared to not controlling for baseline TL.

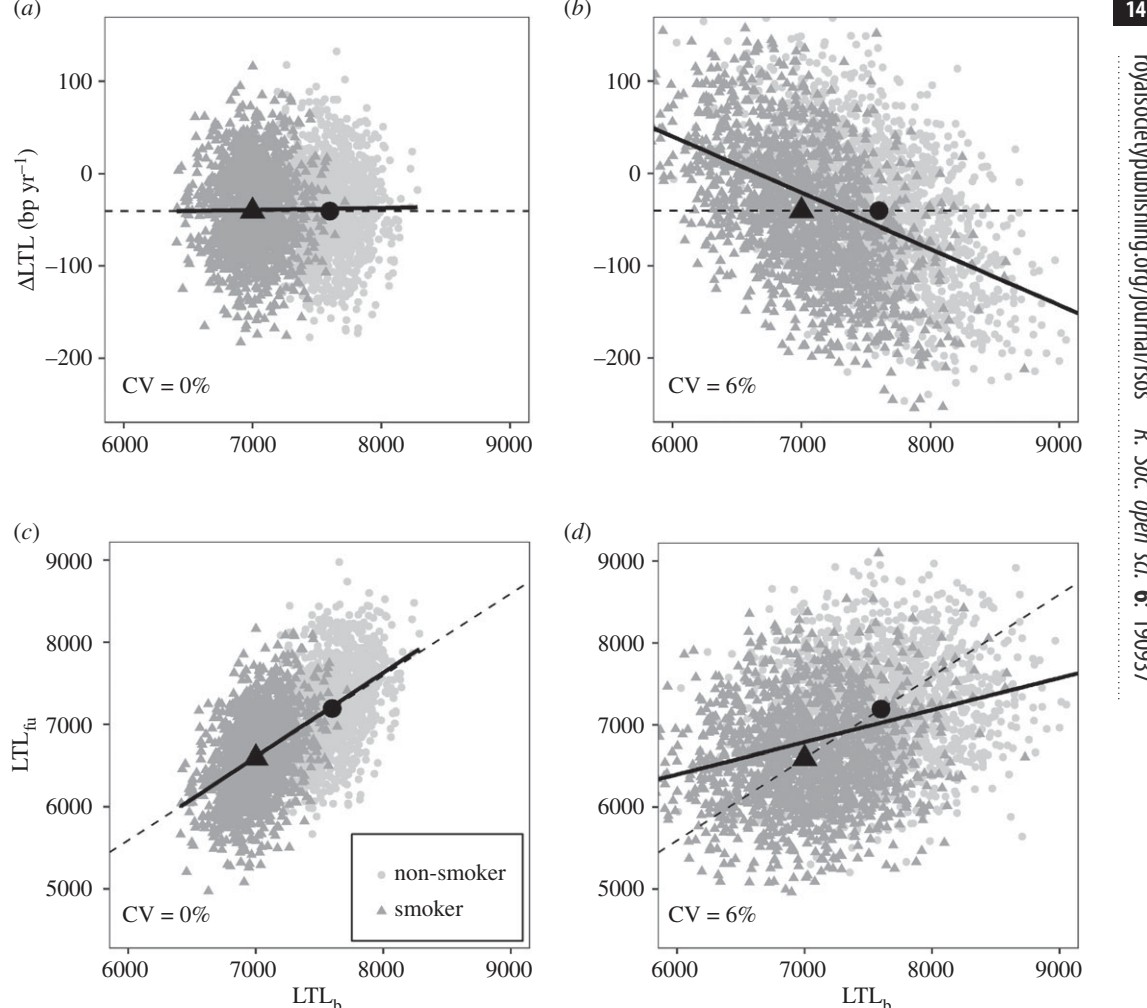

**Figure 7.** Graphical illustration of the biasing effect of controlling for $LTL_b$ in analyses of $\Delta LTL$ and $LTL_{fu}$. This figure is based on simulated data and exaggerates the true difference in LTL between smokers and non-smokers. See text for explanation.

## 4. General discussion

We have used three separate lines of evidence to argue that controlling for $LTL_b$ in analyses of $\Delta LTL$ by adding it to models as a covariate biases estimates of the effects of exposures such as smoking. First, we used directed acyclic graphs to show that under a realistic set of assumptions, $LTL_b$ is likely to be a collider on the path linking smoking and $\Delta LTL$. Controlling for $LTL_b$ is therefore predicted to introduce collider bias in the form of an overestimation of the true difference in $\Delta LTL$ between smokers and non-smokers. Second, we used a simple simulation model to confirm, again under a realistic set of assumptions, that controlling for $LTL_b$ does indeed inflate estimates of the true difference in $\Delta LTL$ between smokers and non-smokers, but only when a true difference in LTL is present at baseline. The magnitude of this bias is positively related to the magnitude of TL measurement error and the presence of bias is not eliminated by correcting for regression the mean resulting from measurement error. Third, we analysed data from seven longitudinal human cohorts and showed that, in line with our predictions, estimates of the difference in telomere attrition between smokers and non-smokers tended to be greater when $LTL_b$ was included in statistical models as a covariate. Furthermore, the magnitude of this latter difference was predicted by proxies for LTL measurement error, as would be expected if the difference arises from collider bias.

Initially, we found it difficult to obtain an intuitive understanding of why controlling for $LTL_b$ is problematic. Figure 7 is an attempt to provide a graphical explanation based on simulated data. The dark grey triangles and pale grey circles indicate LTL measurements for smokers and non-smokers, respectively; the black triangles and circles are the means of the data for smokers and non-smokers,

respectively. All four panels depict LTL measurements from a scenario in which there is a true difference in $LTL_b$ between smokers and non-smokers, but no true difference in $\Delta LTL$ (i.e. scenario C in our simulations). The left-hand two panels (figure 7a,c) show LTL measurements made without error (CV = 0%), whereas the right-hand two panels (figure 7b,d) show the same true LTL values depicted on the left, but now measured with error (CV = 6%). All four panels plot $LTL_b$ on the x-axis, hence in all panels the mean $LTL_b$ for smokers (black triangle) is to the left of the mean $LTL_b$ for non-smokers (black circle). Figure 7a,b plots $\Delta LTL$ as the outcome variable and thus relates to a model 2-type analysis, whereas figure 7c,d plots $LTL_{fu}$ as the outcome variable and thus relates to a model 3-type analysis.

Figure 7a,b shows the association between $LTL_b$ and $\Delta LTL$ as a solid black regression line. When there is no measurement error (figure 7a), there is no relationship between $LTL_b$ and $\Delta LTL$ (the slope is zero). However, when LTL measurement error is introduced (figure 7b), a negative relationship between $LTL_b$ and $\Delta LTL$ occurs as a result of regression to the mean. Controlling for $LTL_b$ in an analysis of the association between smoking and $\Delta LTL$ means asking what the difference in $\Delta LTL$ between smokers and non-smokers is for a given value of $LTL_b$; this is conceptually equivalent to comparing the residuals from the regression of $\Delta LTL$ on $LTL_b$ for smokers and non-smokers (the black line). In figure 7a, the residuals of the data from the regression line are identical for smokers and non-smokers, because the means for smokers and non-smokers lie on the line. However, in figure 7b, the mean for smokers lies below the line, whereas the mean for non-smokers lies above the line. Hence, in figure 7b, residuals are on average negative for smokers and positive for non-smokers creating a spurious difference in the residual $\Delta LTL$ between smokers and non-smokers. This bias only occurs because the smokers have a mean $LTL_b$ that is lower than that of non-smokers; it would not occur if there was no difference in $LTL_b$, because the black triangle and circle would then be in the same place. Figure 7c,d shows the association between $LTL_b$ and $LTL_{fu}$ as a solid black regression line. When there is no measurement error (figure 7c), the slope of the relationship between $LTL_b$ and $LTL_{fu}$ is 1. However, when LTL measurement error is introduced (figure 7d), a flatter relationship between $LTL_b$ and $LTL_{fu}$ results. Controlling for baseline $LTL_b$ in an analysis of the association between smoking and $LTL_{fu}$ causes a spurious difference in $LTL_{fu}$ between smokers and non-smokers in figure 7d via an exactly analogous mechanism to that described for figure 7b.

Given first, that there are robust differences in $LTL_b$ between smokers and non-smokers [2,3], second, that LTL measurement error is often substantial ([34] and figure 5) and third, that most published analyses of the effect of smoking on $\Delta LTL$ or $LTL_{fu}$ control for $LTL_b$, we suggest that the difference in $\Delta LTL$ between smokers and non-smokers is likely to have been overestimated in the literature. Reports of significantly accelerated LTL attrition in smokers compared to non-smokers should therefore be interpreted with caution (e.g. [16,18,28]). In a recent meta-analysis in which we re-analysed LTL data from 18 longitudinal cohorts without control for $LTL_b$, we found no evidence to support accelerated LTL attrition in adult smokers [3]. It is therefore likely that there is in fact no true difference in $\Delta LTL$ between smokers and non-smokers and that an alternative explanation needs to be sought for the robust difference reported in $LTL_b$ [3,15].

Our findings are likely to have much broader implications than the specific case of the effect of smoking on $\Delta LTL$ analysed here. The bias we describe is relevant to estimating the effect of any factor that is associated with a true difference in TL at the time of baseline measurement on the rate of subsequent TL attrition. Indeed, our own analyses suggest that published analyses of the effects of sex and body mass index on $\Delta LTL$ are likely to be biased (see electronic supplementary material). There is a growing literature based predominantly on cross-sectional data claiming that exposure to various forms of stress and adversity accelerates TL attrition [1,51–60]. While cross-sectional associations between exposure to stress and short TL do not prove that stress causes TL attrition [15], longitudinal studies have started to emerge that appear to support a causal relationship [20,22,42]. Unfortunately, just as in the literature on effects of smoking, it is typical for analyses to control for $TL_b$ in these latter studies, meaning that the results should be treated with caution. Re-analyses of these datasets are required to establish whether the claimed differences in TL attrition are in due to bias. We predict that removing $TL_b$ as a control variable from the models used to analyse these data will not just increase the standard error of the estimates (as would be true if $TL_b$ was an innocuous incidental variable that needs to be controlled for to increase power), but will systematically shift the parameter estimates for the effect of the exposure on TL attrition towards zero.

Thus far, we have restricted our discussion to scenarios in which baseline TL is shorter in smokers and/or the rate of telomere attrition is faster in smokers. We focused on these scenarios due to the common assumption that cross-sectional differences in TL are caused by differences in telomere

attrition [15]. However, if this assumption is incorrect, then it is possible that there could be scenarios in which the baseline difference in TL is in one direction and the true difference in the rate of attrition is in the other direction. For example, baseline TL could be shorter in smokers, but the true rate of attrition could be slower. We have simulated such a scenario, and show that the bias produced by controlling for baseline TL eliminates the true difference in attrition at moderate levels of measurement error (CV = ~4%) and reverses it at higher levels of measurement error (electronic supplementary material, figure S12). Thus, baseline control not only exaggerates effects of exposures on attrition, but can also eliminate or reverse them. A recent study provides an example of a dataset in which controlling for baseline TL reverses the estimated effects of age and sex on the rate of telomere attrition, highlighting the importance of properly considering the consequences of baseline control [61].

As a final point, it is worth stressing that our findings are also relevant to areas of epidemiology outside of telomere biology and apply to the analysis of any similarly structured observational studies in which changes over time in imperfectly measured variables are examined. While this problem is understood by some epidemiologists (e.g. [31]), we hope that the current paper raises awareness of measurement error-induced collider bias more widely.

# 5. Conclusion

Controlling statistically for baseline TL by adding it to models as a covariate incorrectly inflates estimates of the difference in telomere attrition between smokers and non-smokers, and the size of this bias is positively related to the size of telomere measurement error. This bias is not restricted to smoking and will occur for any factor that, like smoking, is associated with a systematic difference in TL at the time of the baseline measurement. We found no scenarios in which baseline control yields higher statistical power for detecting true differences in telomere attrition. We therefore recommend that models of telomere attrition should not control for baseline TL by including it as a covariate. Given that the majority of previous analyses of factors affecting telomere attrition control for baseline TL in this way, many claims of accelerated telomere attrition in individuals that are male, older, fatter or exposed to various forms of adversity could be false-positive results that need to be re-assessed.

Data accessibility. An R script that implements the simulations described and an example dataset used to generate the figures in the paper are available at doi:10.5281/zenodo.1009086). Individual participant-level data for CCS, HAS, LBC1921, LBC1936 and NSHD are available on request to bona fide researchers. For NSHD, see http://www.nshd.mrc.ac.uk/data; doi:10.5522/NSHD/Q101 and doi:10.5522/NSHD/Q102.

Authors' contributions. All authors conceived the study, edited the manuscript and approved the final version for publication. M.B. did the simulations, analysed the data and wrote the first draft.

Competing interests. The authors declare no competing interests.

Funding. This work was supported by the National Centre for the Replacement Refinement & Reduction of Animals in Research (grant no. NC/K000802/1 to M.B.), the European Research Council (grant no. AdG 666669, COMSTAR to D.N.) and the United States National Science Foundation (grant no. BCS-1519110 to D.T.A.E.). We thank the following individuals for granting us permission to re-use data from their longitudinal cohorts: Athanase Benetos (ADELAHYDE and Evolution de la Rigidité Artérielle), Yoav Ben-Shlomo (Caerphilly Cohort Study), Cyrus Cooper (Hertfordshire Ageing Study), Diana Kuh (MRC National Survey of Health and Development), Ian Deary (Lothian Birth Cohorts 1921 and 1936). Cohorts acknowledge the following sources of funding: LBC1921, CCS, HAS and NSHD—New Dynamics of Ageing via the HALCyon cross-cohort collaborative programme (RES-353-25-0001); LBC1921—UK Biotechnology and Biological Sciences Research Council (BBSRC), The Royal Society and The Chief Scientist Office of the Scottish Government, Lifelong Health and Wellbeing Initiative (MR/K026992/1); LBC1936—Age UK (Disconnected Mind Project); NSHD—UK Medical Research Council.

Acknowledgements. We are extremely grateful to Jelle Boonekamp and three anonymous referees for helpful feedback on earlier drafts of this paper. We are particularly grateful to one referee for independently confirming the results of our simulations.

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
