## [Reviewer comments · Royal Society Open Science]

Review History

RSOS-190937.R0 (Original submission)

Review form: Reviewer 1

Is the manuscript scientifically sound in its present form?

Yes

Are the interpretations and conclusions justified by the results?

Yes

Is the language acceptable?

No

Do you have any ethical concerns with this paper?

No

Have you any concerns about statistical analyses in this paper?

No

Recommendation?

Major revision is needed (please make suggestions in comments)

Comments to the Author(s)

Dear Editor,

Dear Authors,

Thank you for letting review the manuscript entitled: 'Controlling for baseline telomere length biases estimates of the rate of telomere attrition'. I have read the manuscript with interest. While, I admit it is a topic that enjoys my personal interest, the many empiricists do not know which best statistical model to use and many papers get published with poorly performing statistical models. This certainly is the case here for the problem the authors have chosen. How to best analyse data with this structure (one baseline and one follow-up measurement of the same trait of an individual) received quite some attention a long time ago, it is often insufficiently explained in statistics manuals and the statistical models used are often wrong as the authors show and as has been shown in some references below. I believe the author's study, with its simulations and meta-analysis offer a good addition to the already available literature, especially making the problem and its solution understandable to empiricist, which can be difficult with more statistical manuscripts (like Kronmal 1993, see below). Below I suggest a couple of improvements. First, in a few sections the writing is lengthy and a bit messy. The manuscript would benefit from a sharper, more efficient and more structured writing. Second, in my opinion, the authors could take stronger stand against the poorly performing statistical models and especially clarify from the abstract onwards which is the best performing statistical model (basically, explain model 4 and the increased flexibility of this approach) and that the most common statistical model used in this field overestimates the telomere change.

Specific comments:

Please use your own line numbers.

P2 112 Bias could refer to many things. Could you be more specific?

P2 120 and instead use a model with....explain the model here (model 4). Could you be a bit sharper here?

P2 121 I agree here, but because you used the term bias above, the importance of that message does not get completely through. The problem is that the coefficient was overestimated and that less increased type 1 error. I am not sure you need to explain that already in the abstract, but weighing more the biases will help howing the importance of your manuscript.

P3 136 analytic strategies = statistical models

P5 139-45 This section could be written more efficiently

P5 155 & 158 typos exercise (e.g. 40).

P5 156 interesting, model 2 is actually wrong. This was already shown in Kronmal 1993 (https://www.jstor.org/stable/2983064?seq=1#page_scan_tab_contents). Personally, I would like to motivate the authors should take a stronger negative attitude against it. This is especially important since it is the most common way statistical model in most papers. The issue arises when regressing something that is a function of something else because both are correlated (Kronmal 1992).

P6 l8 I believe Model 4 does contain a correction of mLTlB through the random intercept of individual identity?

P7 l56 Could you clarify the difference between accuracy and precision of parameter estimates?

P7-8 Simulations, results section: I was wondering if you could number your headings and make subsections in this section. This section would from a little more structure. Did I understand it correctly that in this section all the paragraphs following p7 l56 refer to accuracy and precision of parameters estimates?

P8 paragraph l36-50: this is a good summary of your results.

P9 paragraph l6-11: this is a good conclusion paragraph. An additional advantage of model 4 is that it allows a more flexible correction for covariates, e.g. if timepoints 1 and 2 include a range of ages.

P9 l46-53 In this section of the methods, you are giving predictions. It would be clearer if those predictions were not in the methods.

P10 l33-34 Could you specify on average how much bigger (mean, sd)?

P11 l11-14 This is a good illustration of the importance of the problem. If you gave the value of the difference between the model coefficients in the results section, that would give a good feel of the importance of your results.

General comment on the results section: You have two result parts, each one with a results and a discussion section. Then at the end you have a general discussion. I believe it would be helpful if the results and discussion of the simulation were merged. It would also be helpful if the results and discussion of the meta-analysis were merged. This approach would directly link the result with the relevance of the result.

P12 l34-39 This interesting result from your study in reference 3 challenges many earlier studies and this manuscript provides one explanation. Perhaps, you could put more emphasis on that in the motivation of your study and already mention it in the introduction?

Reference:

Kronmal RA. Spurious correlation and the fallacy of the ratio standard revisited. J R Stat Soc A 1993; 156: 379-92.

Review form: Reviewer 2 (Jelle J. Boonekamp)

Is the manuscript scientifically sound in its present form?

Yes

Are the interpretations and conclusions justified by the results?

Yes

Is the language acceptable?

Yes

Do you have any ethical concerns with this paper?

No

Have you any concerns about statistical analyses in this paper?

No

Recommendation?

Accept with minor revision (please list in comments)

Comments to the Author(s)

This study addresses statistical methodological issues relating to the analysis of environmental effects on the rate of telomere attrition. In my opinion the ms was clearly written and the conclusions of this study will be important for anyone analyzing longitudinal data (e.g. repeated measurements) as the identified issues are not limited to telomeres. I have a few suggestions that hopefully will be useful to further improve this ms.

Page 4 Line 17-18: I fully agree. Existing experimental studies on this subject could be cited here to drive home this point.

Page 6 Line 24: To what extent is the “collider bias” similar as “regression to the mean”, since they are both driven by measurement error? At this point in the ms, I am slightly unsure about what this study will add to the existing methodological approaches that deal with regression to the mean. It could be useful to highlight the differences here.

Page 9 line 10: The likelihood of type-1 error driven by measurement error and the inclusion of baseline LTL also depends on the sample size. This must be true because p-values are a direct function of sample size and hence an effect of smoking driven by this bias must become more significant with increasing sample size. Perhaps this is mentioned elsewhere, but I would explicitly mention the dependence of type-1 error on sample size in this section.

Page 9 Line 12-19: This section was not very clear to me. In particular, the underlying reason for why the likelihood of type-1 error increases with sample size in scenario C with model 2 remained unclear. I believe that the reason must be because the p-value is a function of sample size. Perhaps revise this section to include this explanation?

Page 9 Line 12: It might be worth discussing whether a model that includes baseline TL could be advantageous in some situations. Maybe this is not the case in any situation and then it would be worth to mention this. However, I can imagine that when baseline TL does not systematically differ among groups, the inclusion of baseline TL in the model might increase the fit of the model to the data? -- A further point worth mentioning is that a difference in baseline TL among groups in the opposite direction could entirely counter/swamp an effect on subsequent TL. This could be particularly relevant in the situation that multiple groups are to be compared and where a clear a-priori prediction regarding TL differences at baseline cannot be made. Thus, measurement error could drive spurious findings, but at the same time it could also induce type-2 error.

Page 11 Line 22: This section (and fig legend) reads as if a positive correlation between baseline TL and the measured slope of TL attrition reflects measurement error per se. However, there is some (theoretical) evidence suggesting that the rate of telomere attrition depends on baseline TL, with longer telomeres shortening faster, giving rise to the same pattern (e.g. Grasman J, Salomons HM, Verhulst S (2011), Stochastic modeling of length-dependent telomere shortening in *Corvus monedula*. *J. Theor. Biol.* 282, 1–6; Boonekamp, J. J., Simons, M. J., Hemerik, L. and Verhulst, S. (2013), Telomere length behaves as biomarker of somatic redundancy rather than biological age. *Aging Cell*, 12: 330-332). It is therefore not so clear to me that the correlation between initial TL

and subsequent rate of attrition is a good "signature" of measurement error as suggested in Fig 5, even though the variation in slopes among studies suggest that there is indeed substantial variation in measurement error among studies (i.e. a true relation between baseline TL and subsequent rate of attrition should be uniform across studies, assuming that this would be a characteristic feature of a species).

Page 13 Lines 3-26. This is a very useful and clear description of the phenomenon. However, again I wonder to what extent regression to the mean is the same thing. Not that any overlap would make this study less important, but it would be useful to devote a section to link the findings of this study to discuss existing methods dealing with regression to the mean.

Best,

Jelle Boonekamp

Decision letter (RSOS-190937.R0)

31-Jul-2019

Dear Dr Bateson,

The editors assigned to your paper ("Controlling for baseline telomere length biases estimates of the rate of telomere attrition") have now received comments from reviewers. We would like you to revise your paper in accordance with the referee and Associate Editor suggestions which can be found below (not including confidential reports to the Editor). Please note this decision does not guarantee eventual acceptance.

Please submit a copy of your revised paper before 23-Aug-2019. Please note that the revision deadline will expire at 00.00am on this date. If we do not hear from you within this time then it will be assumed that the paper has been withdrawn. In exceptional circumstances, extensions may be possible if agreed with the Editorial Office in advance. We do not allow multiple rounds of revision so we urge you to make every effort to fully address all of the comments at this stage. If deemed necessary by the Editors, your manuscript will be sent back to one or more of the original reviewers for assessment. If the original reviewers are not available, we may invite new reviewers.

- Data accessibility

<http://datadryad.org/submit?journalID=RSOS&manu=RSOS-190937>

- Competing interests

- Authors' contributions

- Acknowledgements

- Funding statement

on behalf of Dr David Wales (Associate Editor) and Kevin Padian (Subject Editor)
openscience@royalsociety.org

Associate Editor's comments (Dr David Wales):

Both reviewers recommend revisions, which they class as major/minor. Please consider all the comments carefully and respond to them in preparing a revised manuscript.

Subject Editor Comments to Author:

We have two good and substantial reviews that are overall encouraging about the manuscript. However they raise a number of points that will need considered attention and so I would like to return the manuscript to the authors to revise and answer the points made. It should not be necessary to send it out for further review if our AE feels that he can look over the changes and responses before publication. Best wishes for your revision and thanks for submitting.

Reviewers' Comments to Author:

Reviewer: 1

Dear Editor,
Dear Authors,

Thank you for letting review the manuscript entitled: 'Controlling for baseline telomere length biases estimates of the rate of telomere attrition'. I have read the manuscript with interest. While, I admit it is a topic that enjoys my personal interest, the many empiricists do not know which best statistical model to use and many papers get published with poorly performing statistical models. This certainly is the case here for the problem the authors have chosen. How to best analyse data with this structure (one baseline and one follow-up measurement of the same trait of an individual) received quite some attention a long time ago, it is often insufficiently explained in statistics manuals and the statistical models used are often wrong as the authors show and as has been shown in some references below. I believe the author's study, with its simulations and meta-analysis offer a good addition to the already available literature, especially making the problem and its solution understandable to empiricist, which can be difficult with more statistical manuscripts (like Kronmal 1993, see below). Below I suggest a couple of improvements. First, in a few sections the writing is lengthy and a bit messy. The manuscript would benefit from a sharper, more efficient and more structured writing. Second, in my opinion, the authors could take stronger stand against the poorly performing statistical models and especially clarify from the abstract onwards which is the best performing statistical model (basically, explain model 4 and the increased flexibility of this approach) and that the most common statistical model used in this field overestimates the telomere change.

Specific comments:

Please use your own line numbers.

P2 l12 Bias could refer to many things. Could you be more specific?

P2 l20 and instead use a model with....explain the model here (model 4). Could you be a bit sharper here?

P2 l21 I agree here, but because you used the term bias above, the importance of that message does not get completely through. The problem is that the coefficient was overestimated and that less increased type 1 error. I am not sure you need to explain that already in the abstract, but weighing more the biases will help howing the importance of your manuscript.

P3 l36 analytic strategies = statistical models

P5 l39-45 This section could be written more efficiently

P5 l55 & l58 typos exercise (e.g. 40).

P5 l56 interesting, model 2 is actually wrong. This was already shown in Kronmal 1993 (https://www.jstor.org/stable/2983064?seq=1#page_scan_tab_contents). Personally, I would like to motivate the authors should take a stronger negative attitude against it. This is especially important since it is the most common way statistical model in most papers. The issue arises when regressing something that is a function of something else because both are correlated (Kronmal 1992).

P6 l8 I believe Model 4 does contain a correction of mLTLb through the random intercept of individual identity?

P7 l56 Could you clarify the difference between accuracy and precision of parameter estimates?

P7-8 Simulations, results section: I was wondering if you could number your headings and make subsections in this section. This section would from a little more structure. Did I understand it correctly that in this section all the paragraphs following p7 l56 refer to accuracy and precision of parameters estimates?

P8 paragraph l36-50: this is a good summary of your results.

P9 paragraph l6-11: this is a good conclusion paragraph. An additional advantage of model 4 is that it allows a more flexible correction for covariates, e.g. if timepoints 1 and 2 include a range of ages.

P9 l46-53 In this section of the methods, you are giving predictions. It would be clearer if those predictions were not in the methods.

P10 l33-34 Could you specify on average how much bigger (mean, sd)?

P11 l11-14 This is a good illustration of the importance of the problem. If you gave the value of the difference between the model coefficients in the results section, that would give a good feel of the importance of your results.

General comment on the results section: You have two result parts, each one with a results and a discussion section. Then at the end you have a general discussion. I believe it would be helpful if

the results and discussion of the simulation were merged. It would also be helpful if the results and discussion of the meta-analysis were merged. This approach would directly link the result with the relevance of the result.

P12 l34-39 This interesting result from your study in reference 3 challenges many earlier studies and this manuscript provides one explanation. Perhaps, you could put more emphasis on that in the motivation of your study and already mention it in the introduction?

Reference:

Kronmal RA. Spurious correlation and the fallacy of the ratio standard revisited. *J R Stat Soc A* 1993; 156: 379–92.

Reviewer: 2

Comments to the Author(s)

This study addresses statistical methodological issues relating to the analysis of environmental effects on the rate of telomere attrition. In my opinion the ms was clearly written and the conclusions of this study will be important for anyone analyzing longitudinal data (e.g. repeated measurements) as the identified issues are not limited to telomeres. I have a few suggestions that hopefully will be useful to further improve this ms.

Page 4 Line 17-18: I fully agree. Existing experimental studies on this subject could be cited here to drive home this point.

Page 6 Line 24: To what extent is the “collider bias” similar as “regression to the mean”, since they are both driven by measurement error? At this point in the ms, I am slightly unsure about what this study will add to the existing methodological approaches that deal with regression to the mean. It could be useful to highlight the differences here.

Page 9 line 10: The likelihood of type-1 error driven by measurement error and the inclusion of baseline LTL also depends on the sample size. This must be true because p-values are a direct function of sample size and hence an effect of smoking driven by this bias must become more significant with increasing sample size. Perhaps this is mentioned elsewhere, but I would explicitly mention the dependence of type-1 error on sample size in this section.

Page 9 Line 12-19: This section was not very clear to me. In particular, the underlying reason for why the likelihood of type-1 error increases with sample size in scenario C with model 2 remained unclear. I believe that the reason must be because the p-value is a function of sample size. Perhaps revise this section to include this explanation?

Page 9 Line 12: It might be worth discussing whether a model that includes baseline TL could be advantageous in some situations. Maybe this is not the case in any situation and then it would be worth to mention this. However, I can imagine that when baseline TL does not systematically differ among groups, the inclusion of baseline TL in the model might increase the fit of the model to the data? -- A further point worth mentioning is that a difference in baseline TL among groups in the opposite direction could entirely counter/swamp an effect on subsequent TL. This could be particularly relevant in the situation that multiple groups are to be compared and where a clear a-priori prediction regarding TL differences at baseline cannot be made. Thus, measurement error could drive spurious findings, but at the same time it could also induce type-2 error.

Page 11 Line 22: This section (and fig legend) reads as if a positive correlation between baseline

TL and the measured slope of TL attrition reflects measurement error per se. However, there is some (theoretical) evidence suggesting that the rate of telomere attrition depends on baseline TL, with longer telomeres shortening faster, giving rise to the same pattern (e.g. Grasman J, Salomons HM, Verhulst S (2011), Stochastic modeling of length-dependent telomere shortening in *Corvus monedula*. *J. Theor. Biol.* 282, 1–6; Boonekamp, J. J., Simons, M. J., Hemerik, L. and Verhulst, S. (2013), Telomere length behaves as biomarker of somatic redundancy rather than biological age. *Aging Cell*, 12: 330-332). It is therefore not so clear to me that the correlation between initial TL and subsequent rate of attrition is a good “signature” of measurement error as suggested in Fig 5, even though the variation in slopes among studies suggest that there is indeed substantial variation in measurement error among studies (i.e. a true relation between baseline TL and subsequent rate of attrition should be uniform across studies, assuming that this would be a characteristic feature of a species).

Page 13 Lines 3-26. This is a very useful and clear description of the phenomenon. However, again I wonder to what extent regression to the mean is the same thing. Not that any overlap would make this study less important, but it would be useful to devote a section to link the findings of this study to discuss existing methods dealing with regression to the mean.

Best,

Jelle Boonekamp

Author's Response to Decision Letter for (RSOS-190937.R0)

See Appendix A.

Decision letter (RSOS-190937.R1)

28-Sep-2019

Dear Dr Bateson,

I am pleased to inform you that your manuscript entitled "Controlling for baseline telomere length biases estimates of the rate of telomere attrition" is now accepted for publication in Royal Society Open Science.

Royal Society Open Science operates under a continuous publication model

(<http://bit.ly/cpFAQ>). Your article will be published straight into the next open issue and this will be the final version of the paper. As such, it can be cited immediately by other researchers. As the issue version of your paper will be the only version to be published I would advise you to check your proofs thoroughly as changes cannot be made once the paper is published.

on behalf of Dr David Wales (Associate Editor) and Kevin Padian (Subject Editor)
openscience@royalsociety.org

Appendix A

Response to referees' comments

Reviewers' Comments to Author:

Reviewer: 1

Thank you for letting review the manuscript entitled: 'Controlling for baseline telomere length biases estimates of the rate of telomere attrition'. I have read the manuscript with interest. While, I admit it is a topic that enjoys my personal interest, the many empiricists do not know which best statistical model to use and many papers get published with poorly performing statistical models. This certainly is the case here for the problem the authors have chosen. How to best analyse data with this structure (one baseline and one follow-up measurement of the same trait of an individual) received quite some attention a long time ago, it is often insufficiently explained in statistics manuals and the statistical models used are often wrong as the authors show and as has been shown in some references below. I believe the author's study, with its simulations and meta-analysis offer a good addition to the already available literature, especially making the problem and its solution understandable to empiricist, which can be difficult with more statistical manuscripts (like Kronmal 1993, see below). Below I suggest a couple of improvements. First, in a few sections the writing is lengthy and a bit messy. The manuscript would benefit from a sharper, more efficient and more structured writing. Second, in my opinion, the authors could take stronger stand against the poorly performing statistical models and especially clarify from the abstract onwards which is the best performing statistical model (basically, explain model 4 and the increased flexibility of this approach) and that the most common statistical model used in this field overestimates the telomere change.

We thank the referee for recognising the value of what we have done. We are not statisticians ourselves and have sought to make a complex topic accessible to other biologists like ourselves who are actively involved in the modelling of telomere data. To clarify the structure of the paper (essentially two separate studies placed between a common introduction and general discussion) we have introduced numbered sections (which we note are supported by RSOS). Although some of the writing is lengthy, we feel strongly that that adequate explanation is necessary to convey what we have done intelligibly to non-statisticians. The positive comments from the second referee (not a statistician) suggest that we have been successful in this attempt and we are therefore reluctant to remove some of the explanation that might seem unnecessary to a statistician.

Specific comments:

Please use your own line numbers.

P2 l12 Bias could refer to many things. Could you be more specific?

We have reworded this sentence more clearly without referring to bias. It now reads: 'Using simulated datasets, we show that controlling for baseline telomere length overestimates of the true effect of smoking on telomere attrition.'

P2 l20 and instead use a model with....explain the model here (model 4). Could you be a bit sharper here?

To clarify what we mean by controlling for baseline telomere length we have rewritten the second sentence of the abstract to be more specific: 'These studies typically control for baseline telomere length by including it as a covariate in statistical models.' We have also rewritten the penultimate sentence of the abstract as follows: 'In conclusion, to avoid invalid inference, models of telomere attrition should not control for baseline TL by including it as a covariate.' We hope that together these two changes make our conclusions sharper. There is really not room in 200 words to go into the models we suggest using instead.

P2 I21 I agree here, but because you used the term bias above, the importance of that message does not get completely through. The problem is that the coefficient was overestimated and that less increased type 1 error. I am not sure you need to explain that already in the abstract, but weighing more the biases will help howing the importance of your manuscript.

By removing any mention of bias from our concluding sentence we hope that we have now clarified our main conclusion.

P3 I36 analytic strategies = statistical models
Indeed...

P5 I39-45 This section could be written more efficiently.

For a statistician, the point we make here is obvious, but for a non-statistician we feel that it is helpful to spell out the advantages of simulation and the overall approach we are taking.

P5 I55 & I58 typos exercise (e.g. 40).
Corrected.

P5 I56 interesting, model 2 is actually wrong. This was already shown in Kronmal 1993 (https://www.jstor.org/stable/2983064?seq=1#page_scan_tab_contents). Personally, I would like to motivate the authors should take a stronger negative attitude against it. This is especially important since it is the most common way statistical model in most papers. The issue arises when regressing something that is a function of something else because both are correlated (Kronmal 1992).

While we fully acknowledge that we are not the first to discuss the general statistical issue we raise (and cite the appropriate references), we are unconvinced that the problem we address is the same problem discussed by Kronmal (1992). Kronmal's paper specifically addresses problems arising from the use of ratios in regression analyses, and none of the models we consider includes ratios as either outcome variables or predictors. Picking up the referee's second point, what they seem to fail to appreciate is that while model 2 includes baseline LTL on both the right and left-hand sides of the model equation (which they are suggesting is wrong), model 3 does not, and yet we show that both models 2 and 3 produce identical, incorrect results. We therefore feel that it is not helpful to cite Kronmal's paper. Despite these disagreements, we are fully in accord with this referee in condemning the use of model 2, and hope this comes across clearly in our conclusions which we have reworded to emphasise our recommendations.

P6 I8 I believe Model 4 does contain a correction of mLTL_b through the random intercept of individual identity?

In the second paragraph of the introduction we say the following: 'Researchers often have a strong intuition that it is important to control for baseline variation in the outcome variable of interest in analyses of change. In the current context, this implies including LTL_b as a covariate (i.e. a continuous predictor variable for which a regression coefficient is estimated) in analyses of the association between smoking and ΔLTL (models 2 and 3 in Table 1).' Following this definition, model 4 does not control for mLTL_b because it does not include mLTL_b as a covariate. We also end the paragraph the referee is referring to here with the clarifying sentence: 'Note that models 1 and 4 contain no control for mLTL_b, in that mLTL_b is not included on the right-hand side of the model equation, whereas models 2 and 3 control for mLTL_b by including it as a covariate and estimating its regression coefficient.'

P7 I56 Could you clarify the difference between accuracy and precision of parameter estimates?

We have deleted this reference to accuracy and precision rather than adding extra words defining the terms.

P7-8 Simulations, results section: I was wondering if you could number your headings and make subsections in this section. This section would from a little more structure. Did I understand it correctly that in this section all the paragraphs following p7 l56 refer to accuracy and precision of parameters estimates?

We have numbered the sections throughout the paper and additionally added some new sub-headings to the simulation results section to make it clearer how the results are organised.

P8 paragraph l36-50: this is a good summary of your results.

Thanks.

P9 paragraph l6-11: this is a good conclusion paragraph. An additional advantage of model 4 is that it allows a more flexible correction for covariates, e.g. if timepoints 1 and 2 include a range of ages.

Thanks. We have noted the additional flexibility of model 4.

P9 l46-53 In this section of the methods, you are giving predictions. It would be clearer if those predictions were not in the methods.

This paragraph is not making predictions, but merely explaining the rationale for our chosen proxies for measurement error. We therefore feel that it is in the correct place in the methods section.

P10 l33-34 Could you specify on average how much bigger (mean, sd)?

We have added the means and sds for the parameter estimates from models 1 and 2 to the text in this paragraph: 'There is a tendency for the coefficients from model 2 to be more negative, indicating a bigger estimated difference in $m\Delta TL \cdot year^{-1}$ compared to model 1 (model 1: mean = 0.046, sd = 0.258; model 2: mean = -0.001, sd = 0.262; paired t-test: $t(6) = 1.87$, $p = 0.1106$). This difference is greater if the comparison is restricted to the five cohorts measured with qPCR (model 1: mean = 0.026, sd = 0.142; model 2: mean = -0.052, sd = 0.158; paired t-test: $t(4) = 3.87$, $p = 0.0180$).'

P11 l11-14 This is a good illustration of the importance of the problem. If you gave the value of the difference between the model coefficients in the results section, that would give a good feel of the importance of your results.

We have now added these values to the results section (see previous response).

General comment on the results section: You have two result parts, each one with a results and a discussion section. Then at the end you have a general discussion. I believe it would be helpful if the results and discussion of the simulation were merged. It would also be helpful if the results and discussion of the meta-analysis were merged. This approach would directly link the result with the relevance of the result.

We have given this suggestion careful consideration, but ultimately decided against it. As the paper is currently structured, the two results sections are divided into sections relating to specific aspects of the results (as requested by this referee), whereas the discussion sections draw out the overall conclusions from each study. It is therefore not easy to see how the results and discussions could be merged without extensive rewriting, which seems dangerous given that the existing text been honed based on the comments of these (and previous) referees.

P12 l34-39 This interesting result from your study in reference 3 challenges many earlier studies and this manuscript provides one explanation. Perhaps, you could put more emphasis on that in the motivation of your study and already mention it in the introduction?

We agree that the bias identified in the current paper is a candidate explanation for why several studies using model 2-type approaches have reported significant effects of smoking on LTL

attrition, whereas our recent meta-analysis using a model 1-type approach found no effect. However, we were not able to directly compare model 1 and model 2 approaches in our published meta-analysis due to not having the raw data available to re-run the models. In the absence of this comparison, the cause of the difference in results is speculative. We do refer to our published meta-analysis in the introduction, but do not feel that it merits a more prominent role in motivating the current paper.

Reviewer: 2

Comments to the Author(s)

This study addresses statistical methodological issues relating to the analysis of environmental effects on the rate of telomere attrition. In my opinion the ms was clearly written and the conclusions of this study will be important for anyone analyzing longitudinal data (e.g. repeated measurements) as the identified issues are not limited to telomeres. I have a few suggestions that hopefully will be useful to further improve this ms.

Thanks for recognising the importance our paper.

Page 4 Line 17-18: I fully agree. Existing experimental studies on this subject could be cited here to drive home this point.

We cite Bendix et al as an example here to show how low the correlations sometime are.

Page 6 Line 24: To what extent is the “collider bias” similar as “regression to the mean”, since they are both driven by measurement error? At this point in the ms, I am slightly unsure about what this study will add to the existing methodological approaches that deal with regression to the mean. It could be useful to highlight the differences here.

Collider bias is driven by regression to the mean arising from measurement error (as we attempt to explain in Figure 7), but it is not the same thing. However, the question of how existing approaches for dealing with regression to the mean affect collider bias is worth asking. To address this question we have added an analysis asking whether collider bias is eliminated by using the correction for regression to the mean suggested by Verhulst et al. We have added this analysis to the simulation section of the paper and included an additional figure (S11) in the Supplementary Material that shows the results. Importantly, using the correction suggested by Verhulst et al does not eliminate bias. We hope that this goes some way to addressing the referee’s question regarding the connections between collider bias and regression to the mean.

Page 9 line 10: The likelihood of type-1 error driven by measurement error and the inclusion of baseline LTL also depends on the sample size. This must be true because p-values are a direct function of sample size and hence an effect of smoking driven by this bias must become more significant with increasing sample size. Perhaps this is mentioned elsewhere, but I would explicitly mention the dependence of type-1 error on sample size in this section.

The referee is absolutely correct that sample size is also important and we make this point in the first paragraph of the discussion of these results: ‘Specifically, the difference in Δ LTL was overestimated and the size of this overestimation increased synergistically with increases in the difference in LTL_b and in LTL measurement error. This bias translated into a type 1 (i.e. false-positive) error rate of above the usually-accepted 5% level when there was no true difference in Δ LTL. This rise in the false-positive error rate was exacerbated in studies with larger numbers of participants due to the positive impact of sample size on power.’

Page 9 Line 12-19: This section was not very clear to me. In particular, the underlying reason for why the likelihood of type-1 error increases with sample size in scenario C with model 2 remained

unclear. I believe that the reason must be because the p-value is a function of sample size. Perhaps revise this section to include this explanation?

The referee is correct in his assumption that the type-1 error rate increases due to the effect of sample size on the p-value. We have rewritten this paragraph to explain this.

Page 9 Line 12: It might be worth discussing whether a model that includes baseline TL could be advantageous in some situations. Maybe this is not the case in any situation and then it would be worth to mention this. However, I can imagine that when baseline TL does not systematically differ among groups, the inclusion of baseline TL in the model might increase the fit of the model to the data?

This intuition is incorrect. Our simulations show that there are no situations in which it is advantageous to include baseline TL by including it as a covariate in the model and our recommendations reflect this. Figure S1 shows that in scenario B (no systematic difference in baseline TL), the power to detect a difference in rates of attrition is identical for models with and without control for baseline TL. The fit of the model with baseline control (model 2) is no better than the model without (model 1), because model 1 includes information about baseline TL by modelling the difference in TL between baseline and follow-up as the outcome variable (rather than by including it as a covariate). We have added the following sentence to the overall conclusions of the paper: “We found no scenarios in which baseline control yields higher statistical power for detecting true differences in telomere attrition.”

A further point worth mentioning is that a difference in baseline TL among groups in the opposite direction could entirely counter/swamp an effect on subsequent TL. This could be particularly relevant in the situation that multiple groups are to be compared and where a clear a-priori prediction regarding TL differences at baseline cannot be made. Thus, measurement error could drive spurious findings, but at the same time it could also induce type-2 error.

This is an extremely important point and is highly relevant given a recent paper showing that the effects of age and sex on the rate of telomere attrition are reversed when baseline TL is added to the models as a covariate. We have added an additional figure to the Supplement (Figure S12) and the following paragraph to the discussion: “Thus far, we have restricted our discussion to scenarios in which baseline TL is shorter in smokers and/or the rate of telomere attrition is faster in smokers. We focussed on these scenarios due to the common assumption that cross-sectional differences in TL are caused by differences in telomere attrition (15). However, if this assumption is incorrect, then it is possible that there could be scenarios in which the baseline difference in TL is in one direction and the true difference in the rate of attrition is in the other direction. For example, baseline TL could be shorter in smokers, but the true rate of attrition could be slower. We have simulated such a scenario, and show that the bias produced by controlling for baseline TL eliminates the true difference in attrition at moderate levels of measurement error (CV = ~4%) and reverses it at higher levels of measurement error (Figure S12). Thus, baseline control not only exaggerates effects of exposures on attrition, but can also eliminate or reverse them. A recent study provides an example of a dataset in which controlling for baseline TL reverses the estimated effects of age and sex on the rate of telomere attrition, highlighting the importance of properly considering the consequences of baseline control (60).”

Page 11 Line 22: This section (and fig legend) reads as if a positive correlation between baseline TL and the measured slope of TL attrition reflects measurement error per se. However, there is some (theoretical) evidence suggesting that the rate of telomere attrition depends on baseline TL, with longer telomeres shortening faster, giving rise to the same pattern (e.g. Grasman J, Salomons HM, Verhulst S (2011), Stochastic modeling of length-dependent telomere shortening in *Corvus monedula*. *J. Theor. Biol.* 282, 1–6; Boonekamp, J. J., Simons, M. J., Hemerik, L. and Verhulst, S. (2013), Telomere length behaves as biomarker of somatic redundancy rather than biological age.

Aging Cell, 12: 330-332). It is therefore not so clear to me that the correlation between initial TL and subsequent rate of attrition is a good “signature” of measurement error as suggested in Fig 5, even though the variation in slopes among studies suggest that there is indeed substantial variation in measurement error among studies (i.e. a true relation between baseline TL and subsequent rate of attrition should be uniform across studies, assuming that this would be a characteristic feature of a species).

We appreciate that there is some evidence to suggest that long telomeres may really shorten faster than short telomeres, and that the correlation between baseline TL and telomere attrition is not just an artefact of regression to the mean. This is why we were at pains to state in our methods section: ‘All else being equal, the correlation between $mLTL_b$ and $mLTL_{fu}$ will be weaker the higher the measurement error, and the correlation between $mLTL_b$ and $m\Delta TL$ will be more negative the higher the measurement error (37,48).’ To further justify our use of these correlations as a proxy for measurement error we have added the following passage to the discussion section: ‘Whilst we appreciate that there is some evidence that long telomeres may really shorten faster (49), there is no reason to expect that any such biological effect will differ between human cohorts. In contrast, there is very good reason to assume that there will be differences in telomere measurement error between human cohort studies. Therefore, it is a reasonable assumption that variation between cohorts in the above correlations reflect variation in measurement error.’

Page 13 Lines 3-26. This is a very useful and clear description of the phenomenon. However, again I wonder to what extent regression to the mean is the same thing. Not that any overlap would make this study less important, but it would be useful to devote a section to link the findings of this study to discuss existing methods dealing with regression to the mean.

We have addressed this comment in a previous response and via the addition of section 2.2.4: Effect of correcting for regression to the mean.